

# Ducks change wintering patterns due to changing climate in the important wintering waters of the Odra River Estuary

Dominik Marchowski[1], Łukasz Jankowiak[1], Dariusz Wysocki[1], Łukasz Ławicki[2] and Józef Girjatowicz[3]

[1] Department of Vertebrate Zoology and Anthropology, Institute for Research on Biodiversity, Faculty of Biology, University of Szczecin, Szczecin, Poland
[2] West-Pomeranian Nature Society, Szczecin, Poland
[3] Hydrography and Water Management Unit, Faculty of Earth Science, University of Szczecin, Szczecin, Poland

Corresponding author
Dominik Marchowski,
marchowskid@gmail.com

## ABSTRACT

Some species of birds react to climate change by reducing the distance they travel during migration. The Odra River Estuary in the Baltic Sea is important for wintering waterfowl and is where we investigated how waterbirds respond to freezing surface waters. The most abundant birds here comprise two ecological groups: bottom-feeders and piscivores. Numbers of all bottom-feeders, but not piscivores, were negatively correlated with the presence of ice. With ongoing global warming, this area is increasing in importance for bottom-feeders and decreasing for piscivores. The maximum range of ice cover in the Baltic Sea has a weak and negative effect on both groups of birds. Five of the seven target species are bottom-feeders (Greater Scaup *Aythya marila*, Tufted Duck *A. fuligula*, Common Pochard *A. ferina*, Common Goldeneye *Bucephala clangula* and Eurasian Coot *Fulica atra*), and two are piscivores (Smew *Mergellus albellus* and Goosander *Mergus merganser*). Local changes at the level of particular species vary for different reasons. A local decline of the Common Pochard may simply be a consequence of its global decline. Climate change is responsible for some of the local changes in the study area, disproportionately favoring some duck species while being detrimental to others.

## INTRODUCTION

Migration distance has declined in several species of aquatic (and other) birds as a result of climate change (*Musil et al., 2011*; *Lehikoinen et al., 2013*; *Meller, 2016*). The distances that birds migrate from their breeding areas in northern and eastern Europe to their central European wintering areas are shorter during mild winters (*Lehikoinen et al., 2013*; *Pavón-Jordan et al., 2015*); conversely, birds may change their wintering sites to warmer regions during colder periods because they may perceive local manifestations of larger scale weather (*Newton, 2008*). Reducing migration distance can provide several benefits

associated with earlier arrival at the breeding grounds and greater survival (*Coppack & Both, 2002*; *Jankowiak et al., 2015a*; *Jankowiak et al., 2015b*). Food resources of wintering sites may also influence migration decisions (*Cresswell, 2014*; *Aharon-Rotman et al., 2016*). Although winter site fidelity is usually very strong among waterfowl (*Newton, 2008*), this can change in response to weather, habitat, and competition (*Cresswell, 2014*). Changing winter sites may often be a trade-off between the costs of finding a new site and the benefits it offers (*Aharon-Rotman et al., 2016*). At sub-zero temperatures, shallow waters freeze over; forcing birds to expend more time and energy searching for food in deeper, open waters. Three functional groups of waterbirds forage in the shallow waters of offshore lagoons: piscivores, herbivores and bottom-feeders, for example, large numbers of waterbirds gather to forage in the Odra River Estuary (hereafter ORE) (*Marchowski et al., 2015*; *Marchowski, Jankowiak & Wysocki, 2016*). Two groups of waterbirds—bottom-feeders and piscivores—are among those most commonly wintering here. During winter, the study area is subject to wide variation in temperatures, often making surface waters subject to freezing (*Van Erden & De Leeuw, 2010*). Yet even relatively small variations in temperature, causing ice cover to form or disappear, can lead to the displacement of waterbirds. Changes in abundance and community structure of birds in the ORE may reflect the impact of climate change. Analysis of the dates of the appearance of ice-related phenomena in the Szczecin Lagoon and of their frequency over time reveals a distinct pattern illustrating recently observed trends in climate warming (*Girjatowicz, 2011*). In this paper, we are looking for how the abundance of some species in the ORE changes due to climate warming. We are investigating whether climate change will differentially influence foraging patterns, and consequently overwintering patterns, of these two groups of birds. Thus, increasing temperatures due to climate change, and the shorter time interval with ice cover will result in increasing numbers of bottom-feeders because they will then migrate shorter distances. Frosts in the study area are never so severe that the water freezes completely to the bottom even in the shallows. But even if ice does cover the surface of shallow water bottom-feeders have no access to food and have to move to warmer areas because feeding areas where sedentary mussels are abundant tend to be in shallow waters (*Marchowski et al., 2015*). Piscivores, on the other hand, can still feed in such conditions (frozen surface of shallows, unfrozen deeper areas - further from the shore, where there are no mussels but there are fish) and remain in the area, e.g., observation of large aggregations of piscivores during harsh winters (Smew *Mergellus albellus* and Goosander *Mergus merganser*) (*Kaliciuk, Oleksiak & Czeraszkiewicz, 2003*; *Czeraszkiewicz, Haferl & Oleksiak, 2004*; *Marchowski & Ławicki, 2011*; *Guentzel et al., 2012*; *Marchowski & Ławicki, 2012*; *Marchowski, Ławicki & Guentzel, 2013*). This has consequences for conservation management plans in protected areas. Two different groups of birds react differently to climate warming, showing different patterns of moving closer to their breeding grounds, as a consequence in our area should be more bottom-feeders. Thus, changes in the importance of wintering ground due to changing surface-water freezing patterns expected under global warming regimes are likely to have important consequences for very large numbers of these birds.

## STUDY AREA

The study area in the south-western Baltic Sea forms the Polish part of the Odra River Estuary system, which includes the Great Lagoon (the Polish part of the Szczecin Lagoon), Świna Backward Delta, Kamień Lagoon, Dziwna Strait and Lake Dąbie (522.58 km$^2$, Fig. 1). The area comprises four interconnected Important Bird Areas (IBA) and also a Natura 2000 area (*Wilk et al., 2010*). The average and maximum depths of the estuary are 3.8 and 8.5 m, respectively; the dredged shipping lane passing through the estuary from the Baltic Sea to the port of Szczecin is 10.5 m deep (*Radziejewska & Schernewski, 2008*). Waters of the Szczecin Lagoon, Kamień Lagoon, and Lake Dąbie are brackish. Salinity in the central part of the estuary varies from 0.3 psu to 4.5 psu (mean = 1.4 psu) and declines with increasing distance from the sea (*Radziejewska & Schernewski, 2008*). Average winter temperature is 0.3 °C (*Weatherbase, 2016*). The ORE is subject to high levels of eutrophication (*Radziejewska & Schernewski, 2008*). Communities of benthic organisms are typical of freshwater bodies, and the fauna includes large populations of zebra mussels *Dreissena polymorpha*, which was introduced in the mid-19th century. By the 1960s, the biomass of zebra mussels in the Szczecin (Great) Lagoon was estimated at 110,000 metric tons (*Wiktor, 1969*; *Wolnomiejski & Woźniczka, 2008*). The distribution of the zebra mussel is extremely uneven (see the map in *Marchowski et al., 2015*). The average density of the zebra mussel in the ORE is 0.18 kg /m$^2$, but the vast majority occupies around 10% of the area, where the mean density is 2.05 kg/m$^2$ (*Stańczykowska, Lewandowski & Czarnoleski, 2010*). Fish are mainly freshwater species such as roach *Rutilus rutilus*, bream *Abramis brama*, pike *Esox lucius*, perch *Perca fluviatilis* and ruff *Gymnocephalus cernua*; there are also anadromous fish including smelt *Osmerus eperlanus* and occasionally herring *Clupea harengus* among others (*Wolnomiejski & Witek, 2013*).

## METHODS

### Bird censusing

The study covers two functional groups of waterbirds: bottom-feeders (diving birds, feeding on motionless type of food—mussels)—Greater Scaup (*Aythya marila*—hereafter Scaup), Tufted Duck (*A. fuligula*), Common Pochard (*A. ferina*— hereafter Pochard), Common Goldeneye (*Bucephala clangula*—hereafter Goldeneye) and Eurasian Coot (*Fulica atra*—hereafter Coot); piscivores (diving birds, feeding on mobile type of food—fish)—Smew and Goosander (*Stempniewicz, 1974*; *Cramp & Simmons, 1977*; *Johansgard, 1978*). Six of our target species belong to the order: Anseriformes, family: Anatidae, subfamily: Anatinae and Tribe: Mergini (Goldeneye, Smew and Goosander), Tribe: Aythyini (Scaup, Pochard and Tufted Duck); one species—Coot belongs to the order: Gruiformes, family: Rallidae (*Del Hoyo & Collar, 2014*). Although Coot is not closely related to the rest of our species, we included it into a common group of waterbirds and, due to its behavior and ecology, to the bottom-feeders group (*Stempniewicz, 1974*; *Taylor & Kirwan, 2016*). These seven species are usually very abundant in the study area (*Kaliciuk, Oleksiak & Czeraszkiewicz, 2003*; *Czeraszkiewicz, Haferland & Oleksiak, 2004*; *Wilk et al., 2010*; *Marchowski & Ławicki, 2011*; *Guentzel et al., 2012*; *Marchowski & Ławicki, 2012*; *Marchowski, Ławicki & Guentzel, 2013*).

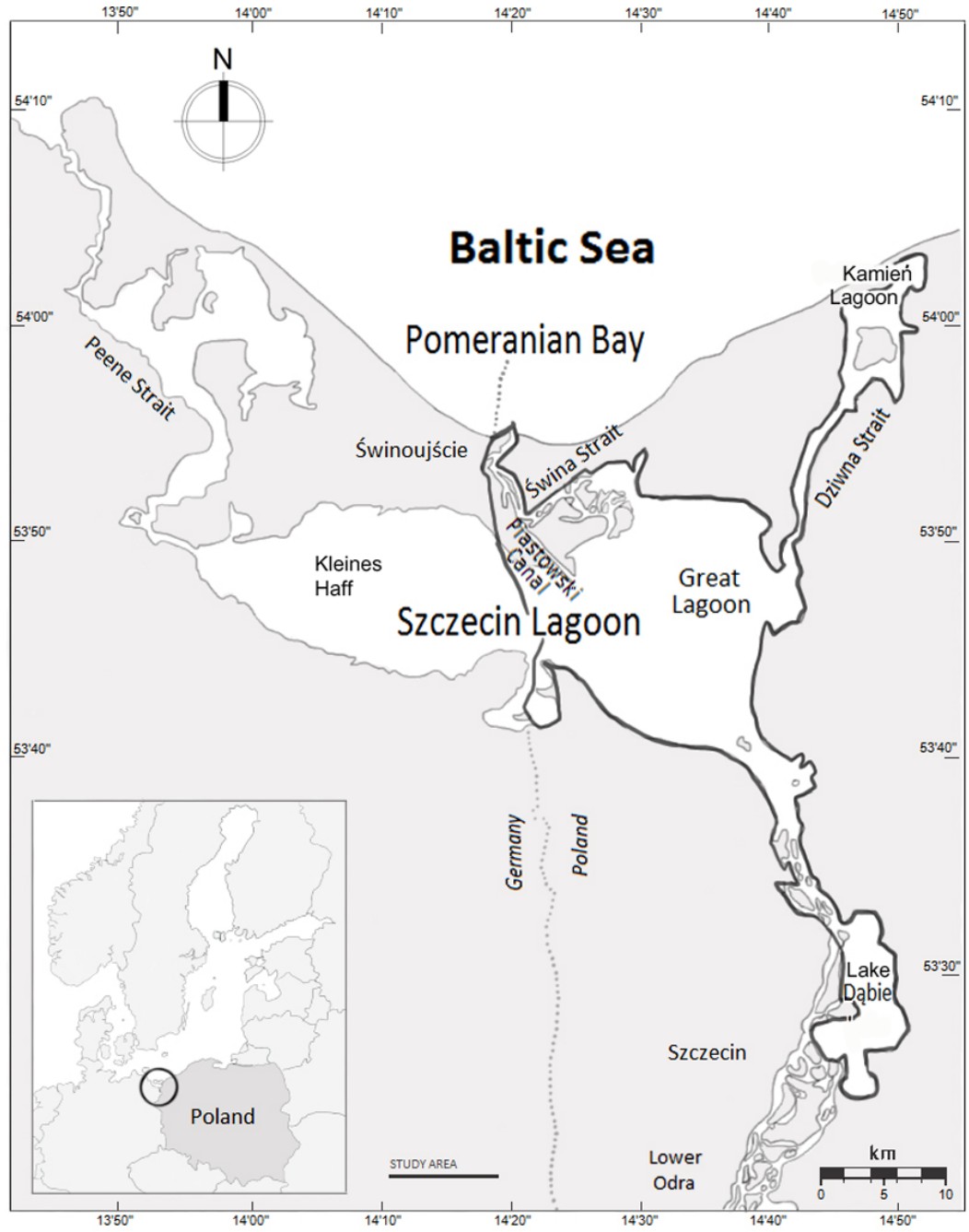

**Figure 1  The Odra River Estuary, north-western Poland.**

These are important gathering areas for populations of these birds associated with regional migration flyways. These subpopulations (hereafter regional or flyway populations) are: Pochard—north-east Europe / north-west Europe; Tufted Duck—north-west Europe (wintering); Scaup—northern Europe / western Europe; Goldeneye—north-west and

central Europe; Smew—north-west and central Europe (wintering); Goosander—north-west and central Europe (wintering); Coot—north-west Europe (wintering) (*Wetlands International, 2016*).

Censuses were conducted using standard methods for non-breeding season waterbirds counts (*Komdeur, Bertelsen & Cracknell, 1992*; *Wetlands International, 2010*). Birds were counted during 17 seasons (1991/1992 to 1993/1994 and 2001/2002 to 2015/2016) during the migration and wintering periods between November and April. Three censuses were carried out per season in November, January, and March or April; there was one midwinter count in January in 2001/2002. Altogether we analyzed the results of 44 counts. Most counts were done on foot. Each observer was equipped with 10x binoculars and tripod-mounted spotting scopes. Observers walked along the same routes, and the same counting method was used during every census every year. Additionally, fourteen aerial counts were made at an average speed of about 100 km/h and an altitude of about 80 m above the water (see Table S1 for the method of data collection: aerial or ground). In the early 1990s, counts were aerial, whereas in 2009–2015 parallel aerial and ground counts were carried out (to compare methods). In ice-free conditions, the species covered in this study can be assigned to a group with just a small error between methods (<6%), one species—Coot had a moderate error (16%), the ground method estimated greater numbers than the aerial one. During periods with more than 70% ice cover, abundance estimated from the air was greater than that estimated from the ground (Dominik Marchowski, pers. comm., 2016). Count method was treated as a random effect in the model. The detailed methodology and results of the counts are given elsewhere (*Meissner & Kozakiewicz, 1992*; *Meissner, Kozakiewicz & Skakuj, 1994*; *Kaliciuk, Oleksiak & Czeraszkiewicz, 2003*; *Czeraszkiewicz, Haferland & Oleksiak, 2004*; *Marchowski & Ławicki, 2011*; *Guentzel et al., 2012*; *Marchowski & Ławicki, 2012*; *Marchowski, Ławicki & Guentzel, 2013*). Where large numbers of unidentified *Aythya* species were counted—26,000 ducks in November 2009, 13,000 in November 2010, 6,000 in January 2012, 3,300 in March 2012 and 13,500 in November 2015—they were estimated to be in the ratio of 1:0.8 (scaup:tufted) based on observations in other studies. Observations were always at a distance that does not disturb the birds in any way, and in Poland, this research required no ethical or scientific permits.

## Statistical analysis

Absolute numbers of birds can vary widely and independently, and so the proportion of the local population size (in our study area) in relation to flyway population for each species was used as our dependent variable (*Nagy, Flink & Langendoen, 2014*; *Wetlands International, 2016*). Thus, if we used the trend of absolute numbers in our area, the resulting error would be the larger, the greater the changes in the size of the flyway population. Therefore, we indicate the numbers of a species by means of a coefficient calculated as the percentage of the flyway population present in the study area during a particular count. We obtained the regional population size estimates from 1992 to 2012 from *Nagy, Flink & Langendoen (2014)*; for the period 2013–2016, we used the flat trend calculated by *Nagy, Flink & Langendoen (2014)* (Table 1). Initially, we placed the

**Table 1** Regional flyway populations and annual trends (after *Nagy, Flink & Langendoen, 2014*) for seven species of waterbirds using the Odra River Estuary.

| Species[a] | Functional group[b] | Number of individuals (1992)[c] | Number of individuals (2012)[d] | Population trend % p.a.[e] | Significance of changes[f] |
|---|---|---|---|---|---|
| Greater Scaup | B | 300 | 150 | −3.57 | Large decline |
| Common Pochard | B | 280 | 150 | −3.35 | Large decline |
| Tufted Duck | B | 1,100 | 820 | −0.98 | Large decline |
| Goosander | P | 130 | 100 | −0.09 | Stable |
| Eurasian Coot | B | 990 | 950 | +0.19 | Moderate increase |
| Common Goldeneye | B | 210 | 240 | +0.26 | Moderate increase |
| Smew | P | 13 | 24 | +1.97 | Large increase |

Notes.
[a] Target species.
[b] Functional group: B, bottom-feeders, P, piscivores.
[c] Estimated number of individuals from regional flyway population in 1992, the numbers are presented in thousands.
[d] Estimated number of individuals from regional flyway population in 2012, the numbers are presented in thousands.
[e] Population trend % per annum - long term assessment.
[f] Significances of changes.

different species in ecological groups. The bottom-feeders (denoted by B) included Scaup, Tufted Duck, Pochard, Goldeneye and Coot, and the piscivores (P) contained Smew and Goosander. We used the minimum temperatures averaged over the 15 days leading up to the count day. The climate data were obtained from the Szczecin weather station (53.395N, 14.6225E, http://tutiempo.net). Another climate covariate was ice cover in the study area; data relating to this were published by the Polish Institute of Meteorology and Water Management. These data are from the observation point at Miroszewo on the shore of the Szczecin Lagoon (53.734N, 14.331E, http://www.imgw.pl/). We compared the number of days with 100% ice cover in the period from 0 to 15 days prior to the bird counts. The ice cover of 100% refers specifically to the Miroszewo observation point. This estimate is a good approximation for the region. In practice, however, the ORE is never completely covered by ice (*Girjatowicz, 1990*; *Girjatowicz, 2005*; see the Discussion for an explanation) and birds are still present in such conditions. We also utilized the maximum ice extent in the Baltic Sea (max ice; data obtained from the website of The European Environment Agency; *EEA-The European Environment Agency, 2017*). Apart from climatic variables, we also tested the changes in species occurrence during the survey years, so we used the season as a covariate. Prior to the final analysis, we checked multicollinearity between the above variables using the variation inflation factor (VIF). VIFs of all variables were in acceptable limits, minimum temperatures (VIF = 2.1), max ice (VIF = 1.03), ice cover (VIF = 2.07) and season (VIF = 1.04). However, we found a moderate linear significant relationship between minimum temperature and ice cover ($r = 0.52$, $p < 0.001$) and after exclusion of minimal temperature VIF showed no multicollinearity issue between variables—ice cover (VIF = 1.04), max ice (VIF = 1.03), season (VIF = 1.03)—and these were used in the subsequent analyses. Frozen water by definition impedes bottom-feeding duck foraging, we are testing whether that pattern is changing in association with climate change. We

used a general linear mixed model (GLMM) to test above-described relationship. The percentage of the flyway population present in the study area, estimated by species, was used as a target variable using the normal distribution response distribution and identity link function. Mixed models permitted repetition across survey months, methods (aerial and ground counts) and species (random intercept). Thus, to test our assumptions we included the following interactions: feeding group*season, feeding group*ice cover and feeding group*max ice. Selection of the best model structure for the dependent variable was based on the Akaike information criterion (AIC) (*Zuur et al., 2009*). All possible models were carried out (they are listed in Table S2 in Supplementary material). As the final models, we assumed those in which $\Delta$AIC <2 (*Burnham & Anderson, 2002*) and in our case, it was only a general model with all the tested variables. To test the relationship between explanatory variables and particular species abundance we performed for each species GLMM model, where random effects were month and method. We checked also the relationship between winter year and ice cover using a generalized linear model with negative binomial error distribution. We used IBM SPSS Statistic version 20 software for the statistical analysis. $P < 0.05$ was considered statistically significant.

## RESULTS

Below, we present the results of our analyzes, the main one jointly examines two whole groups: piscivores and bottom-feeders and the second one where each species was analyzed. Bird numbers by feeding group were different in their relationships with ice cover; bottom-feeding species in the study area were more sensitive to lower temperatures and left sooner when colder weather increased ice cover, whereas a number of piscivorous species had a higher tolerance to the extent of ice cover (Table 2). Interactions between feeding group and season, feeding group and ice cover, and feeding group and maximum ice extent on the Baltic sea were all important (Table 2). However, the strongest effects were interactions with ice cover, then interaction with the season; this translates into increases in the importance of the site for bottom-feeders. The effect of maximum ice extent was very small (Table 2). Population indices in the ORE changed over the last 25 years, increased in the case of bottom-feeding species but decreased for piscivorous species (Table 2). Ice cover across the whole Baltic Sea had the same, though weak, impact on both functional groups of birds, numbers of birds in the ORE declined with expanding ice cover in the Baltic (Table 2).

Where particular species are concerned, the situation is more complex. The population indices of Scaup and Tufted Duck in the ORE exhibited an increasing trend ($\beta_{Scaup} = 0.026 \pm 0.010$ s.e., $p = 0.011$; however for Tufted Duck it was marginally insignificant $\beta = 0.008 \pm 0.004$ s.e., $p = 0.058$), despite the general decline in their entire northern and western European populations; numbers of both species in the ORE were adversely affected by ice cover in that region ($\beta_{Scaup} = -0.065 \pm 0.011$ s.e., $p < 0.001$; $\beta_{Tufted} = 0.032 \pm 0.005$ s.e., $p < 0.001$) but not by ice cover in the whole Baltic ($\beta_{Scaup} = -0.099 \pm 0.074$ s.e., $p = 0.186$; $\beta_{Tufted} = 0.010 \pm 0.029$ s.e., $p = 0.735$). Relative numbers of Pochard in the ORE have declined ($\beta = -0.009 \pm 0.003$ s.e., $p = 0.005$), but

![PeerJ]

**Table 2** Results of general linear mixed models for seven species showing the influence of ice cover, maximum ice extent (km²) in the Baltic Sea (max ice) and season on the percentage of occurrence of bottom-feeders (denoted by B, Scaup, Tufted Duck, Pochard, Goldeneye, Coot) and piscivores (denoted by P, Smew, Goosander) in the Odra River Estuary. Species, method and month were treated as random effects.

| Model Term | Coefficient | Std. Error | t | P |
|---|---|---|---|---|
| Intercept | 26.553 | 11.619 | | |
| Ice cover | 0.014 | 0.006 | 2.375 | **0.018** |
| Season | −0.013 | 0.006 | −2.204 | **0.028** |
| Max ice | −0.114 | 0.040 | −2.824 | **0.005** |
| Feed[B] | −38.751 | 11.959 | −3.240 | **0.001** |
| Season*Feed[B] | 0.019 | 0.006 | 3.212 | **0.001** |
| Ice cover*Feed[B] | −0.044 | 0.007 | −6.623 | **<0.001** |
| Max ice*Feed[B] | 0.094 | 0.046 | 2.071 | **0.039** |
| Species (r) | 0.074 | 0.048 | | |
| Method (r) | 0.015 | 0.020 | | |
| Month (r) | 0.001 | 0.002 | | |

so has the whole northern European population, abundance was negatively impacted by ice cover in both the study area ($\beta = -0.015 \pm 0.003$ s.e., $p < 0.001$); and in the entire Baltic ($\beta = -0.053 \pm 0.023$ s.e., $p = 0.023$). For Goldeneye, the index for the ORE population was unchanged ($\beta = -0.004 \pm 0.004$ s.e., $p = 0.275$), despite the increase in the European population, abundance was negatively impacted by ice cover in the study area ($\beta = -0.016 \pm 0.004$ s.e., $p < 0.001$), but not by ice cover in the entire Baltic ($\beta = -0.044 \pm 0.027$ s.e., $p = 0.107$). Relative numbers of Coot in the ORE remained unchanged ($\beta < 0.001 \pm 0.002$ s.e., $p = 0.915$), despite the slight increase in the European population, abundance was negatively impacted by ice cover in both the study area ($\beta = -0.014 \pm 0.003$ s.e., $p < 0.001$), and in the entire Baltic ($\beta = -0.038 \pm 0.016$ s.e., $p = 0.019$). The ORE population index for Smew decreased ($\beta = -0.020 \pm 0.008$ s.e., $p = 0.024$), despite the increase in its flyway population, abundance in the ORE was unaffected by ice cover either in the study area ($\beta = -0.013 \pm 0.010$ s.e., $p = 0.183$), or in the Baltic as a whole ($\beta = -0.079 \pm 0.061$ s.e., $p = 0.204$). Finally, relative numbers of Goosander in the ORE remained unchanged ($\beta = -0.005 \pm 0.008$ s.e., $p = 0.572$), like those of the whole population wintering in north-western and central Europe; abundance in the ORE was unaffected by ice cover either in the study area ($\beta = 0.018 \pm 0.009$ s.e., $p = 0.063$) or in the Baltic as a whole ($\beta = -0.111 \pm 0.057$ s.e., $p = 0.057$). The details relating to all these species are listed in Table 1, presented in Fig. 2A and Table S3. Table 3 summarizes the changes in the importance of the ORE for wintering populations of diving waterbirds in the last 25 years. There was a negative relationship between year and winter ice ($\beta_{intercept} = 82.011$, s.e. $= 28.49$, $\beta_{winter} = -00.40$, s.e. $= 0.0142$, Chi-square $= 8.001$, $df = 1$, $p = 0.005$; Fig. 2B).
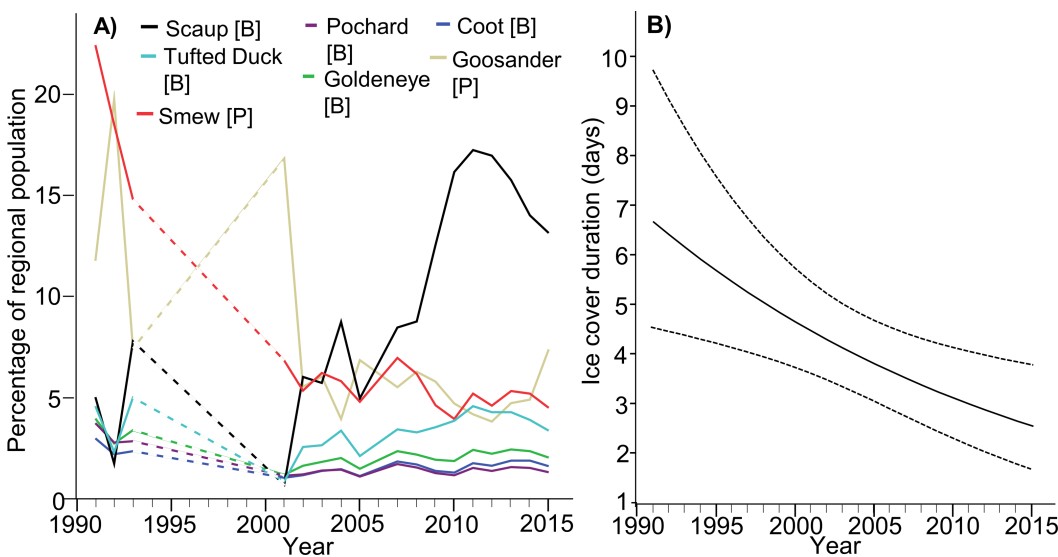

**Figure 2** (A) Predicted results of the general linear mixed model showing the changes of the percentage of the target species population in the Odra River Estuary during years 1992–2016. The predicted values were obtained from the model where we added species as a fixed variable. The model's parameters are listed in **Table S3**. Dashed lines –the gap with birds data. (B) Changes in the ice cover duration in the Odra River Estuary during years 1992–2016. Results of a generalized linear model (with negative binomial error distribution) - correlation between year and winter ice. Dotted lines show 95% confidence intervals bounds.

**Table 3** Population index trends in the Odra River Estuary (ORE) for the regional biogeographic (flyway) population (b.p.) of diving waterbirds showing the percentage of the flyway population in 1992; the percentage of the flyway population in 2016; the mean percentage of the flyway population in the period 1992–2016 ± standard error; and the trend in the period 1992–2016.

| Species | %b.p.1992 | %b.p.2016 | Mean1992–2016 ± SE | Trend in ORE |
|---|---|---|---|---|
| Greater Scaup | 5.68 | 12.60 | 14.17 ± 2.84 | ↑ |
| Tufted Duck | 2.87 | 4.79 | 2.61 ± 0.25 | ↑ |
| Common Goldeneye | 4.48 | 0.63 | 1.21 ± 0.14 | → |
| Eurasian Coot | 0.86 | 0.68 | 0.61 ± 0.07 | → |
| Goosander | 12.59 | 1.80 | 6.85 ± 1.01 | → |
| Smew | 7.04 | 2.76 | 7.01 ± 1.27 | ↓ |
| Common Pochard | 1.84 | 0.20 | 0.62 ± 0.09 | ↓ |

## DISCUSSION

The phenomenon of freezing in our study area has decreased over time (*Girjatowicz, 2011*, Fig. 2B), so that target birds species should tend to feed more recently more often than in the past. However, two functional groups of waterbirds—bottom-feeders and piscivores—react differently to ice cover, a factor that is directly connected to climate change; this has consequences for the wintering patterns of these species. Bottom-feeding birds (Scaup, Tufted Duck, Pochard, Goldeneye, and Coot) tend to be more sensitive to ice cover in the study area than piscivores (Smew and Goosander). Piscivores can survive in colder areas, closer to their breeding ranges, but bottom-feeders have to move further

south and west. This phenomenon indicates that piscivores are declining in our study site because they are shifting further north and east in order to stay closer to their breeding areas. Bottom-feeders are increasing their number for the same reason—they, too, are moving further north and east—but in their case, the result is a greater number in our study area and a smaller one in areas further west and south. Bottom-feeding birds forage in the ORE mainly on mussels of the genus *Dreissena* (*Marchowski et al., 2015*; *Marchowski, Jankowiak & Wysocki, 2016*); the highest quality of this food resource is found in water 1–2 m deep (*Wolnomiejski & Witek, 2013*). Shallow water freezes over faster, displacing birds to deeper unfrozen areas where food is accessible only with difficulty. In addition, when ice cover is present, the abundance of food in unfrozen areas declines owing to its greater exploitation, because the birds congregate on a limited area. In the case of piscivorous birds, increasing ice cover does not affect their numbers negatively. The ORE is never completely covered by ice: the shipping lane between Świnoujście and Szczecin is kept free of ice (*Girjatowicz, 1990*; *Girjatowicz, 2005*), and there are always other areas free of ice, especially at the mouths of the small rivers flowing into the estuary. These ice-free areas may still abound in fish and provide food for piscivores. In general, there is growing importance of the study area for the bottom-feeders. The main analysis jointly examines two whole groups—piscivores and bottom-feeders; the results of it are the positive effect of not freezing for bottom-feeders and negative effect for piscivores, and also a positive effect of the season for bottom-feeders, and negative effect for piscivores (see Table 2). In the other analysis, which examined each species separately, there are differences between them (see Fig. 2A). As far as the most numerous species—Scaup and Tufted Duck—are concerned, the results tally with those of the first analysis, i.e., their numbers are increasing (correlation with season). For the less abundant species, the result of the second analysis is different: Pochard numbers, for example, are decreasing (negative effect). In the first analysis, abundance and season were negatively correlated in piscivores. In the second analysis, in which we examined species separately, abundance and season were independent in the Goosander, while abundance and season were negatively correlated in the Smew. In fact, the strongest effect of the first years of the study relates to the piscivores, especially Smew. If we take the entire period (1992–2016), the Smew declines in numbers (confer with model results), which was due to their being very abundant in the early 1990s. From 2002–2016, Smew numbers are stable.

An interesting result is the negative effect of maximum ice cover in the entire Baltic Sea on the numbers of all species in our study. This is unexpected since our study area is in the warmer south-western Baltic, where one would anticipate an increase in the number of waterbirds in such circumstances (*Alerstam, 1990*). The explanation for this relationship is not easy and certainly goes far beyond the scope of this work, but we can speculate on possible scenarios. On maps with the maximum range of ice cover in the Baltic Sea we can see clearly that when the northern Baltic, i.e., the Gulf of Bothnia and the Gulf of Finland, is completely frozen over, the entire Pomeranian Bay (SW Baltic) (see the map—Fig. 1) together with the ORE is also covered with ice (*Finnish Meteorological Institute, 2017*). These areas freeze over quickly because of their shallowness and low salinity, the latter being due to the considerable influence of fresh water from the Odra river basin. Consequently, during harsh winters, birds from northern Baltic move to the south and west, but they by-pass our

study area as it is covered by ice. Under such circumstances, there may sometimes be better conditions for waterbirds in areas farther north, e.g., the southern coast of Sweden, where there is no ice cover (*Finnish Meteorological Institute, 2017*). Worth noting here, however, is that such cold weather causing the entire Pomeranian Bay and Odra River Estuary to freeze over is rare and becoming rarer (*EEA-The European Environment Agency, 2017*).

The global temperature has risen about 1 °C over the last 130 years, and Northern Hemisphere temperatures of the last 30 years have been the highest in over 800 years (*Stocker et al., 2013*). The extent and duration of ice cover in the Baltic have decreased on average by 50% over the last 36 years (*Schröder, 2015*). There is evidence that the range and occurrence of migratory birds have changed in response to climate change and that some species have shortened their migratory movements by wintering closer to their breeding areas (*Musil et al., 2011*; *Lehikoinen et al., 2013*; *Pavón-Jordan et al., 2015*; *Meller, 2016*). Assuming continued climate warming, the negative correlation between numbers of bottom-feeding birds and the number of days with ice cover indicates that the ORE is becoming more important for this group of birds. Climate change seems to be the primary reason for increases (in the study area) in numbers of Scaup and Tufted Duck and decreases in numbers of Smew; this corresponds with the findings of *Lehikoinen et al. (2013)* in the case of Tufted Duck and of *Pavón-Jordan et al. (2015)* in the case of Smew. Our results are important for conservation planning. Declines in the flyway populations of species such as Scaup and Tufted Duck, even though the importance of our study area to these species is increasing, but at the same time, there is an increase in exposure to locally emerging threats. The biggest threats to these species in the area include fishery bycatches (*Žydelis et al., 2009*; *Bellebaum et al., 2012*). The ecology of diving birds makes this type of threat responsible for the extra mortality of all species covered by this study. Comparison of a species' estimated total population numbers (*Nagy, Flink & Langendoen, 2014*) with numbers for the ORE is interesting, since local trends and European trends do not always concur. The different responses of particular species to the factors investigated are also worth examining. We grouped the species by trends in the study area and discuss these for each species below.

## Species with increasing population index in the study area

Between the late 1980s and 2012, the population of Scaup wintering in northern and western Europe declined at an annual rate of 3.57%/year (*Nagy, Flink & Langendoen, 2014*). Around 41% of the Scaup from this population spent the winter in the Baltic Sea region (*Skov et al., 2011*), and this, in turn, declined by 60% from 1991 to 2010 (*Aunins et al., 2013*). At the same time, we found that the importance of the ORE for this species was increasing. Scaup numbers increased by 300% in the Szczecin Lagoon (the biggest part of Odra River Estuary—see the map—Fig. 1) and the eastern coastal areas of Germany, as opposed to declines further west along the German coast, where some areas (Wismar Bay and Traveförde) had fewer birds than 15 years earlier (*Skov et al., 2011*). A similar trend was found in Sweden, where the number of wintering Scaup increased between 1971 and 2015 (*Nilsson & Haas, 2016*). But farther west, in the Netherlands, *Hornman et al. (2012)* recorded decreases at the most important wintering sites since 1980/1981. All of these

studies confirm that Scaup is shifting its wintering range northwards and eastwards, closer to its breeding areas: this is the reason for the heightened importance to this species of the ORE, even as its overall population wintering in northern and western Europe is declining.

Tufted Duck populations wintering in north-western Europe have recently been decreasing by 0.98%/year (*Nagy, Flink & Langendoen, 2014*). The population estimated for the North-West Europe flyway remained relatively stable between 1987 and 2009 (*Lehikoinen et al., 2013*), a situation confirmed by *Wetlands International (2016)*. In the Baltic Sea region, too, there were no significant changes in numbers between 1991 and 2010 (*Aunins et al., 2013*). We have found that our study area has increased in importance for this species, although not to the same extent as for Scaup. Swedish populations, by comparison, have increased between 1971 and 2015 (*Nilsson & Haas, 2016*), and *Lehikoinen et al. (2013)* reported a rapid increase in the last three decades for Finland. Tufted Ducks in the ORE behave in the same way as Scaup in that they form mixed flocks consuming the same type of food (*Marchowski, Jankowiak & Wysocki, 2016*). At a larger scale, Tufted Ducks have a different migration and wintering strategy: Scaup concentrate in a few hot spots, moving jump-wise between them, whereas the distribution of Tufted Ducks is more diffuse (*Van Erden & De Leeuw, 2010*; *Skov et al., 2011*; *Neubauer et al., 2015*; *Carboneras & Kirwan, 2016a*; *Carboneras & Kirwan, 2016b*). This could cause Tufted Ducks to disperse to smaller water bodies outside our study area, e.g., the numerous lakes in the Pomeranian Lake District in northern Poland ($\sim$34,000 km$^2$), whereas Scaup remains almost exclusively in the ORE (e.g., *Marchowski & Ławicki, 2011*; *Marchowski, Ławicki & Guentzel, 2013*). Scaup is known to concentrate in big flocks during migration and wintering, and the whole flyway population may be concentrated in a few hot-spots such the ORE (*De Leeuw, 1999*; *Marchowski et al., 2015*): this is important in the context of species conservation planning. There is an increase in the importance of the ORE for Scaup, but at the same time, there is an increase in exposure to locally emerging threats such as bycatches in fishing nets (*Bellebaum et al., 2012*). Taking into account the above pattern of Scaup behavior and our results, there is a justified fear that locally operating threats in the ORE may affect the entire flyway population of the species. This is one of the most important messages of our work.

## Species with decreasing population index in the study area

Pochard populations from north-east/north-west Europe have declined rapidly at an annual rate of 3.35%/year (*Nagy, Flink & Langendoen, 2014*). Pochard numbers in the Baltic Sea region also declined by 70% between 1991 and 2010 (*Aunins et al., 2013*). In 1995 there were an estimated 300,000 Pochard in the north-east/north-west European population (*Delany et al., 1999*). With a constant decline of 3.35%/year, the total population should now be less than 150,000 (*Nagy, Flink & Langendoen, 2014*). Numbers of Pochard were expected to be higher in the ORE because of the reduced ice cover. However, we found a reduction in the importance of the estuary to this species (Table 3), corresponding with its global decline (*Aunins et al., 2013*; *Nagy, Flink & Langendoen, 2014*; *BirdLife International, 2015a*; *BirdLife International, 2015b*; *Wetlands International, 2016*). Pochard behaves more like Tufted Duck than Scaup over winter in being more dispersed and occurring on smaller bodies of water (e.g., *Marchowski & Ławicki, 2011*; *Marchowski, Ławicki & Guentzel, 2013*;

*Neubauer et al., 2015*). This implies that individuals may also be wintering outside the study area, e.g., on the numerous water bodies of the Pomeranian Lake District, like Tufted Duck. This local decline, however, seems to be driven by the species' global decline, despite the emergence of better conditions for wintering that might favor population growth.

Smew populations wintering in northern, western and central Europe increased at 1.97%/year between the late 1980s and 2012 (*Nagy, Flink & Langendoen, 2014*); in the Baltic Sea region, numbers increased by 30% between 1991 and 2010 (*Aunins et al., 2013*). Although Smew cannot be classified as a piscivore in the same way as Goosander (and Red-breasted Merganser *M. serrator*), it does feed on very small fish and on small invertebrates (*Carboneras & Kirwan, 2016c*). Though more dependent on shallow water than Goosander, Smew generally forages on mobile types of food. So even if shallow waters freeze over, it may remain on site and search for food in deeper water, which is what we have observed. We found that today, the ORE is of less importance to Smew (Table 3) as illustrated by the northward and eastward shift in wintering area boundaries due to climate warming (*Pavón-Jordan et al., 2015*; *Nilsson & Haas, 2016*).

### Species with no changes in the population index in the study area

Coot populations wintering in north-west Europe increased by 0.19%/year between the late 1980s and 2012 (*Nagy, Flink & Langendoen, 2014*), but in the Baltic region, there was a 60% decline between 1991 and 2010 (*Aunins et al., 2013*). We have found no changes in Coot numbers in the ORE over the last 25 years (Table 3). Likewise, no changes in numbers were recorded between 1975 and 2010 at wintering sites in warmer areas to the south-west (the Netherlands) (*Hornman et al., 2012*). Long-term figures for Sweden (1971–2015), while not revealing any distinct increase, do show that Coot populations fluctuated, rising during mild periods and falling during cold periods (*Nilsson & Haas, 2016*). The expected increase in numbers due to improvements in habitat quality did not happen. Factors such as pressure from American mink *Neovison vison*, which is responsible for the decline of Coot in many places (e.g., *Ferreras & Macdonald, 1999*), may have held back potential increases. Moreover, compared to the bottom-diving Ducks, Coot is more sensitive to cold weather: a study by *Fredrickson (1969)* demonstrated high mortality after periods of severe weather (also reflected in the results of Swedish breeding bird surveys—Leif Nilsson, pers. comm., 2017) but that the population recovered during mild winters. This factor may also be the reason for the different reactions of Coot and diving ducks to the cold.

Goldeneye populations wintering in north-west and central Europe increased at 0.26%/year between the late 1980s and 2012 (*Nagy, Flink & Langendoen, 2014*) and increased in the Baltic Sea region by 50% between 1991 and 2010 (*Aunins et al., 2013*). This corresponds to the data provided by *Lehikoinen et al. (2013)*, an increase in numbers in the northern Baltic wintering area (Finland and N Sweden), but a decline in the southern part of its wintering range (Switzerland, France). In our work, we found the relative number of Goldeneye in the ORE to be stable in the period 1992–2016 (Table 3). This again tallies with the findings of *Lehikoinen et al. (2013)* that duck abundances are independent of temperature in the central part of the flyway. This is probably why the shift in wintering range is not perceptible in our study area but is more pronounced at other, e.g., Swedish

wintering sites, where numbers have increased (*Nilsson & Haas, 2016*) but not in the Netherlands, where they have declined (*Hornman et al., 2012*).

Goosander populations wintering in north-west and central Europe have been stable since the early 1990s (*Nagy, Flink & Langendoen, 2014*); moreover, numbers in the Baltic Sea between 1991 and 2010 did not change significantly (*Aunins et al., 2013*); we also found non-significant changes in the ORE.(Table 3). As in the case of Goldeneye, the explanation is that in the central part of the flyway, species abundances are independent of temperature. In other areas, observations indicate a shift farther to the north and east in the wintering range as a result of climate warming (*Hornman et al., 2012*; *Lehikoinen et al., 2013*; *Nilsson & Haas, 2016*).

## CONCLUSION

Climate change can influence the distribution of overwintering waterbirds. Apart from climate changes, however, feeding ecology, interspecific competition, fishery and other human-related disturbance may be also important and should be taken into consideration (*Quan, Wen & Yang, 2002*; *Žydelis et al., 2009*; *Clavero, Villero & Brotons, 2011*; *Eglington & Pearce-Higgins, 2012*). Protected areas covered by our study will be more important for some species (Scaup and Tufted Duck) but less so for others (Smew). Taking into account the large abundance of the target species regularly present in the ORE, conservation measures applied here will have a large impact on entire populations and will be particularly important for Scaup. Shifts in species distributions should be accounted for in future management plans for Special Protection Areas of the European Natura 2000 network. Our results add new insight to the problem of wintering waterbirds protection and can help to shape conservation policy in the southern Baltic.

## ACKNOWLEDGEMENTS

We thank all the people who took part in the fieldwork—mainly members of the West-Pomeranian Nature Society—but especially the most active among them during the entire study period: Michał Barcz, Ryszard Czeraszkiewicz, Sebastian Guentzel, Michał Jasiński, Zbigniew Kajzer, Jacek Kaliciuk, Krzysztof Kordowski, Aneta Kozłowska, Wojciech Mrugowski, Arkadiusz Oleksiak, Bartosz Racławski, Tomasz Rek, Artur Staszewski, Marcin Sołowiej, Piotr Siuda, Paweł Stańczak and Maciej Przybysz. We are grateful to Leif Nilsson, James Roper and the two anonymous reviewers for their valuable comments on the first version of the manuscript, and to Peter Senn, who kindly improved our English.

### Funding

Birds were counted by volunteer members of the West-Pomeranian Nature Society. Since 2011, mid-winter counts are part of a larger program called Monitoring of Birds of Poland, a project commissioned by the General Inspectorate of Environment Conservation (GIOŚ), supported by the National Fund for Environmental Protection and Water Management to

(NFOŚiGW), and coordinated by the Polish Society for the Protection of Birds (OTOP) http://monitoringptakow.gios.gov.pl/about-project. The funders had no role in study design, data collection and analysis, decision to publish, or preparation of the manuscript.

### Grant Disclosures
The following grant information was disclosed by the authors:
National Fund for Environmental Protection and Water Management.
Polish Society for the Protection of Birds (OTOP).

### Competing Interests
The authors declare there are no competing interests. Łukasz Ławicki is a member of the West-Pomeranian Nature Society, and all his work with this article was contributed non-profit as a volunteer.

### Author Contributions
- Dominik Marchowski and Łukasz Jankowiak conceived and designed the experiments, performed the experiments, analyzed the data, contributed reagents/materials/analysis tools, wrote the paper, prepared figures and/or tables, reviewed drafts of the paper.
- Dariusz Wysocki conceived and designed the experiments, performed the experiments, analyzed the data, contributed reagents/materials/analysis tools, wrote the paper, reviewed drafts of the paper.
- Łukasz Ławicki and Józef Girjatowicz conceived and designed the experiments, performed the experiments, contributed reagents/materials/analysis tools, wrote the paper, reviewed drafts of the paper.

### Animal Ethics
The following information was supplied relating to ethical approvals (i.e., approving body and any reference numbers):

The research consisted of observations of birds from a distance - such studies do not cause disturbance of birds. In Poland, such studies do not need special permission or approval.

### Data Availability
The raw data has been supplied as a Supplementary File.

### Supplemental Information
Supplemental information for this article can be found online at http://dx.doi.org/10.7717/peerj.3604#supplemental-information.

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
