# Peer review of "Ducks change wintering patterns due to changing climate in the important wintering waters of the Odra River Estuary"

_PeerJ, doi:10.7717/peerj.3604_

## Round 0.1 · original submission · Major Revisions

Three reviewers have now commented on your manuscript. Based on the advice received, I suggest you to revise your manuscript according to the review reports.

Reviewer 1 ·

Basic reporting

The manuscript of Marchowski et al deals about wintering waterbird numbers in one important wetland. Although the topic is interesting, there are several issues that need to be clarified before the manuscript is good enough to be published. Especially the methodology has been poorly described and thus the reader can not evaluate has it been done appropriately. I have also suggested some alternative ways to do the modelling. The language would also benefit from a check of a native speaker.

Experimental design

Based on this text it is not clear to the reader what was the model structure. Could you give the model formula or explain it more detailed. Since you apparently have multiple testing (one species at time), you should use Bonferroni correction or similar to adjust P-values. Instead of modelling one species at time, you could also consider multispecies model where feeding type is a covariate and species is a random factor. You could also consider some more general climate variable, like the maximum ice cover of the Baltic Sea, which could tell situation outside your study site, since this will influence also the wintering numbers in Odra.

Validity of the findings

Based on this text it is not clear to the reader what was the model structure. Could you give the model formula or explain it more detailed. Since you apparently have multiple testing (one species at time), you should use Bonferroni correction or similar to adjust P-values. Instead of modelling one species at time, you could also consider multispecies model where feeding type is a covariate and species is a random factor.

Additional comments

The manuscript of Marchowski et al deals about wintering waterbird numbers in one important wetland. Although the topic is interesting, there are several issues that need to be clarified before the manuscript is good enough to be published. Especially the methodology has been poorly described and thus the reader can not evaluate has it been done appropriately. I have also suggested some alternative ways to do the modelling. The language would also benefit from a check of a native speaker. I hope you find these useful.


Abstract

L22-26 The logic is here odd. Climate change is occurring even though the birds would not response. Therefore we do not need birds to confirm the direction of the climate change. However, birds can response to climate change and that is what you are investigating. Please rephrase.

Introduction

L61-> You could introduce the study species in the methods part

L78-85 Here the reader starts to wonder: “But if the whole area will get froze then there won’t be any habitat for piscivorous species either”. Please add already here that the whole area is never frozen. You should also tell somewhere in the introduction or methods how the ice conditions have developed. Is the a long term trend, which why you could expect increasing waterbird numbers. If the ice situation has been stable, then there is no reason to expect changes in waterbird numbers on climatic point-of-view.

L111 It is not mentioned has the methodology remained the same during the study period and if not, how it has changed and how this could have influenced results.

L152 You probably mean that there are always some wintering bird numbers in the area, but the numbers can vary depending on ice conditions. You could also consider some more general climate variable, like the maximum ice cover of the Baltic Sea, which could tell situation outside your study site, since this will influence also the wintering numbers in Odra.

L160-167 Based on this text it is not clear to the reader what was the model structure. Could you give the model formula or explain it more detailed. Since you apparently have multiple testing (one species at time), you should use Bonferroni correction or similar to adjust P-values. Instead of modelling one species at time, you could also consider multispecies model where feeding type is a covariate and species is a random factor.

L169-171 This has been written more like discussion style, consider rewriting so that you only tell the results.

L172-> The methodology of this has not been mentioned. Please add this to the methods. It is now not possible to evaluate is the methodology appropriate.

L195 “and our results supported our assumption.”

L231-> “This could cause that tufted ducks may disperse…

Species-specific discussion. Please mention what is the area of the populations that you are discussing. This is not obvious in most of the cases.

·

Basic reporting

Maybe someone should overlook the English used. As I have not English as my mother-tounge I am not prepared to go into details.

There is one reference missing in the reference list and there is at least one study that should be cited and discussed as it catches important Points. I have put this in the spoecific comments.

Experimental design

No comment

Validity of the findings

No comments

Additional comments

The paper is of great interest but I have a number of specific Points listed in the attachment that should be considered when working furhter with the manuscript.

Reviewer 3 ·

Basic reporting

General
The title of the paper, “Birds in estuaries can act as indicators of climate change: a study at a key site for waterbirds in Europe” is ambitious. I fully see the point of the idea. Nevertheless the paper does not provide any suggestions as to how these data can actively be used as indicators for climate change.
Most studies of the impact of climate change on bird distributions are based on bird data and climate data from a wide range of locations. See for instance Lehikoinen et al. 2013 in Global Change Biology and Pavon-Jord et al, 2015 in Diversity and Distributions.
This paper bases their conclusions on data from one particular site, and this weakens the conclusions.

There could be three ways to meet this:

1. To re-title the paper and describe changes in waterbird numbers as a response to climate change
2. To give more precise suggestions as to how precisely these birds could be used as indicators
3. To target the paper on fewer species, for instance Scaup and the piscivoros species, and widen the geographical range to more sites along the wintering range of the species..
There are numerous comments to the manus, but I expect the paper will undergo a major change, and thus not relevant at this stage.

Language
The English language of this paper needs a thorough polish. Both in terms of sentence building and in terms of extra words that has not been erased. The extent of this is beyond the task of this present review.

Experimental design

The study has the benefit of a long time series of waterbird counts, which is impressive.
The bird count data consist of a combination of land-based surveys and aerial surveys. A column in the primary data table on survey platform would be useful. A thorough discussion of the comparability of those results would be desirable.

The collection of data on ice-cover is impressive. Yet, the methods section describes data originating from the Polish Institute of Meteorology and data collected by the authors, collecting data on a number of factors at an unspecified number of stations. One observation point is mentioned in particular from where 75 % of the lagoon can be observed “on a clear day”. The statistical analysis methods section it is indicated that mainly data from that one observation point was used in the analysis, and not the data from the Polish Institute of Meteorology. This needs to be clarified.

In the primary data table two columns of ice cover information is given, one by the name of “ice15” and one by the name of “ice 0-1”. I assume that ice15 will be the number of days with 100 % ice cover prior to a survey, as mentioned in the text, but there is no mention of the “ice 0-1” data. This could well be whether or not 100 % ice cover at the day of the survey. This needs clarification.

It would be very interesting to collect data on ground temperature as an alternative to the ice cover variable. A temperature variable could be treated as cumulated mean daily temperature at a set number of days prior to a bird count, and provide less categoric information than the “100 % ice cover” covariate. And such a covariate would also benefit from being less subjective. And actually, in line 170 of the Results part, the authors use the term “temperatures” as a proxy for the ice cover.

Validity of the findings

References to for instance Lehikoinen et al. 2013 would have been relevant.

The definition of the variable "ice cover" should be considered exchanged or supplemented with temperature data.

Conclusions on Scaup, Smew and Goosander seems robust and less clear on the remaining species. A more narrow species selection could be considered, or alternatively one of the approaches suggested under "Basic reporting".

Additional comments

For general comments, please see "Basic reporting"

Specific comments

L. 99-100:
In many cases eutrophic situations are beneficial to bivalves. Are you sure that the term “adverse effects” is appropriate here?
L. 125-129:
Is the ratio between Scaup and Tufted Duck very variable between count and is it variable with season. This would be important to elucidate.

---

## Round 0.2 · Major Revisions

I have included most of my comments and observations within the attached PDF of your manuscript. You will find many observations on writing style and I think you could easily reduce the text by writing more clearly and succinctly. There are a few points I should emphasize.

1) You seem to emphasize some trivial aspects of your study. For example, if the water is frozen to the bottom, then benthic feeders cannot feed. That is not an interesting result, but rather it is a logical consequence of feeding in this environment. The interesting result is that the number of days available FOR FEEDING has changed over time. THAT should be emphasized. You did a similar logic for the piscivores. But really, you can show that piscivores can feed even under surface ice layers and so ice should not be as important.

2) You use the word "assume" when you really should have used the word "predict" - but it is not clear that this would always be the perfect word in context. Please consider this.

3) You use the word "significantly" with two different meanings. One is "importantly" and the other really is "significantly." I prefer to not use the word. After all, if a statistic IS significant, then you can state the direction of the pattern, while if it is not, then there is no direction to the "pattern." So, simply by stating the direction, the word significant is implied. When the pattern is important, then the word should be important. So, that only comes after analyzing the results. A statistically significant result can be unimportant. For example, the next point.

4) You state very small percentage changes in population sizes - how can we know that they are important biologically? THAT is the question you should be emphasizing.

5) in your discussion, you separate the species into units for discussion. I don't think you need to. You should group the species by trends. That will make the text more succinct and will emphasize your point better.

I did not comment on the entire discussion and conclusion because I felt that with all my other comments, you should recognize where you can improve the writing style. Also, the discussion will have to change a bit if you follow my other comments elsewhere. You will note in the PDF that I have many points that will make the text more clear and succinct.

You should also re-read the comments of the previous reviewers to insure that you resolved their questions and comments.

Finally, I would change the title. Something along the lines of "Water birds in the Baltic Sea are changing wintering patterns due to climate change." or something along those lines.

I hope you find these comments useful and constructive.

· Appeal

Appeal


· · Academic Editor

Reject

After considering the all comments from the three reviewers in first round processing and the comments from the reviewer 3 in second round processing, I decided to reject your manuscript for publication in PeerJ.

Reviewer 3 ·

Basic reporting

I do struggle to find a good angle to this manus. The title is ambitious. But it doesn’t take the discussion to a level that meets the ambitions. The title could be “Relationship between numbers of selected waterbird species wintering in the ORE area and ice cover”.

There are a number of examples where sentences are not readable, or just not finalized to an expected level.

The discussion is lacking a wider perspective to the issue of impact of climate change to bird wintering distribution.

Experimental design

The analysis of the relationship between abundance of a number of waterbird species with ice cover is well performed, but I am lacking discussion of whether this is purely a climate change effect or if other explanations could potentially be in play.

The authors have considered to use temperature covariates in addition to ice cover covariates. They exclude the use of temperature covariates with reference to high VIF. And define a weak threshold of "around 2". They do not provide the VIF values for temperature covariates in a table and neither do they do so for the other covariates used.

Validity of the findings

See attached text.

Additional comments

Review of ”How do waterbirds respond to climate change? A study at a key wintering site in Europe. (#15149)

I do struggle to find a good angle to this manus. The title is ambitious. But it doesn’t take the discussion to a level that meets the ambitions. The title could be “Relationship between numbers of selected waterbird species wintering in the ORE area and ice cover”.
For the climate covariates, three values of temperature was tested, but then disregarded with reference to high VIF-value.

L176…185
Here you describe to methods of deriving covariates on ice cover for the study site. You quote that you use the variable from the “Monthly Ice Listing”, but why then describe how you collected local data on ice cover every day at noon. I can’t see those data utilized in the tables.

L. 194…
If VIFs were well above 2, the relevant variables were excluded from the analysis. Hence, we excluded the mean, maximum and minimal temperatures averaged over the 15 days prior to the count, as they were highly correlated.

The wording “well above..” is a very loose term. You must come up with an exact threshold, and the VIF value should be given in the relevant tables for all covariates. And it would be good to see which of the covariate sets (on ice and on temperature) that provide the best fit with bird abundances.

Moreover, a number of examples of incomplete sentences appear in the text, indicating that the manuscript has not been lifted to the necessary level.
For instance

L.226…229
The indices for Scaup and Tufted Duck increased in the ORE, despite the general decline in the numbers of species wintering in northern and western Europe, the negative impact of ice cover in the study area on abundance and the lack of any impact of ice cover on the Baltic Sea.
which is unreadable.

I assume that it should be:
The indices for Scaup and Tufted Duck increased in the ORE, despite the general decline in the numbers of species wintering in northern and western Europe, and despite the negative impact of ice cover in the study area on abundance and the lack of any impact of ice cover on the Baltic Sea.
Moreover, in the same sentence you are not talking about “number of species” but rather “number of individuals by species”.

The indices for Scaup and Tufted Duck increased in the ORE, despite the general decline in the numbers of species wintering in northern and western Europe, the negative impact of ice cover in the study area on abundance and the lack of any impact of ice cover on the Baltic Sea.

L. 159-160:
The dependent variable was the percentage of occurrence of a given species in relation to the total estimated population size in a given year.
I do not think that you can talk about an annual population estimate. The figures in Nagy et al. are total figures from the annual Mid-winter counts coordinated by Wetlands International, and these cannot be regarded annual population estimates, but rather annual totals from the coordinated counts. This goes for all the places in the text where referred to Nagy et al.

L. 244 … 245
As we had predicted, benthic feeding birds (Scaup, Tufted Duck, Pochard, Goldeneye and Coot) were more sensitive to the presence of ice cover in the study area.
You need to define the comaprison:
As we had predicted, benthic feeding birds (Scaup, Tufted Duck, Pochard, Goldeneye and Coot) were more sensitive than … to the presence of ice cover in the study area.

L. 249-250
In addition, the food richness of unfrozen areas is declining owing to their greater exploitation.
This sentence needs a lot of better explanation. As it stands it doesn’t make sense.

L. 271…273
Consequently, the birds move to the south and west when there is thick ice cover in the northern Baltic, but they probably by-pass our study area.
The statement “probably” is rather weak and need further elaboration.

In a lot of the citings of Ainars et al. 2013 the “et al.” part has been forgotten.

---

## Round 0.3 · Major Revisions

While I appreciate your efforts to improve the manuscript, I still feel that it has several issues that need to be addressed. I think in my previous review and comments, you might have misinterpreted some of my questions. I assumed that when I asked a question, you would recognize that the text was unclear on that point, and you would then re-write the text. However, sometimes in your rebuttal letter, you only answered my question to me, without changing the manuscript. Here I have several more comments and include your current manuscript PDF with comments and suggestions.

First, the title. I think you could make it shorter and more to-the-point. Thus, something along the lines of: Ducks change wintering patterns due to changing climate in the important wintering waters of the Odra River Estuary. (As I said, something like that if not exactly that).

Next, you are presenting your study as if it were a test of hypotheses, when in fact, it is a description of patterns (important still!). That is, frozen water by definition impedes bottom-feeding duck foraging - it's not something you test, but rather is something you can show. What you CAN test is whether that pattern is changing IN ASSOCIATION WITH climate change. I would really like to see the "hypothesis" testing part worked on a bit more. Also, in one place you suggested that piscivores should have the inverse pattern of the bottom-feeders, when in fact, piscivores should be independent of these freezing patterns.

Your analyses seem to be fine, but they should be interpreted correctly with respect to your hypotheses, which need work.

Also, because you are saying that your patterns are due to climate change, I would think that in your figures could be more informative. For example, instead of having three, you could simply show lines for Ice Cover and Maximum ice extent in the first panel by year. After all, these three figures are of the same patterns.

If you did something along those lines, you could enlarge the first panel and illustrate the causes all within the first graph. Also, instead of Season as the X axis in A, I would use Year (and write the entire year).

Along those same lines, I would think you could take your idea to the next step and include some kind of prediction of (under current rates of climate change) say the next 10 years and if freezing is reduced, how large the effect is predicted to be on the ducks.

Please read my first review as well as this one to insure that you have clarified all the points I was making.

---

## Round 0.4 · Minor Revisions

I hate to be a curmudgeon, but I still have some concerns about the manuscript. There are two main things of importance that are still sort of stumbling blocks. One, in the abstract and conclusions, you seem to be making a case for a NEGATIVE effect on the smew while a POSITIVE effect on the bottom-feeding ducks. However, there is no reason or logic to think that NOT freezing matters for the smew. I think you overstate your case - but if not, you need to make it more explicit.

Next, in looking again af the figures, it looks like all species but the scaup have more or less constant population estimates since 2002. Also, there seems to be little or no relationship between their populations and that of the ice cover duration. Thus, the ice cover duration figure does not APPEAR to support your ideas. In summary, it looks like all the trends are due to what happened BEFORE 2002 and not since then. If this is true (and it would be easy to test), it seems to be a nearly fatal flaw in the analyses. Can you clarify all that?

---

## Round 0.5 · Minor Revisions

I know this might be frustrating, but this time, the changes are really very minor, but I felt that the text should be fixed as much as possible before accepting. Here are the details that need changing.

In the abstract especially, but elsewhere in the text, you used the word "showed" too often, and not always grammatically correctly. In general, you use the verb "to show" which is passive voice, when you could have simply stated the trend. Here is an example:

Your text: "In the case of piscivores the first analysis showed the negative correlation between season and abundance among the fish-eaters. In the second analysis, where we assessed each species separately, Goosander showed no change (no correlation between season and abundance) whereas Smew did reveal a change (there was a correlation between season and abundance, tending towards a reduction in numbers).

My suggestion: "In the first analysis, abundance and season were negatively correlated in piscivores. In the second analysis, in which we examined species separately, abundance and season were independent in the Goosander, while abundance and season were negatively correlated in the Smew."

Our sentences tell the exact same story. Note that I simply eliminated the word "showed" and wrote the sentences more directly. I told the same story in 40 words that you did with 59 words. You should do a search for the word "showed" and see if you can substitute a different way of writing as in my example here. Please try to be more direct and succinct.

Next, you need to standardize "piscivores" and eliminate "fish-eaters." If you prefer the word "benthivores" you may standardize it as well, but I usually find the word "bottom-feeders" for these ducks to occur more often in the literature. Take your pick, but standardize.

Minor details include the word "whole" when the word "entire" would be more appropriate.

In line 287-289, you should clarify that you are talking about the Smew. Note that the sentence that ends with Smew (line 286) ends with a comma and not a period, as it should. The sentence that follows should be - "If we take the entire period (1992 - 2016), the Smew declines in numbers (confer with model results), which was due to their being very abundant in the early 1990s. From 2002 - 2016, Smew numbers are stable."

Note that I used "we" less, and wrote more directly about the organisms.

Please use these examples as models and try to review the rest of the text for similar types of grammar. You will note that my style is more succinct, but without losing information or readability.

---

## Round 0.6 · accepted · Accept

I appreciate your work in improving the style and clarity of your manuscript. This version is considerably better-written than the first version, so I think the work has been worthwhile.